# Aging Behavior and Heat Treatment for Room-Temperature CO-Sensitive Pd-SnO_2_ Composite Nanoceramics

**DOI:** 10.3390/ma15041367

**Published:** 2022-02-12

**Authors:** Fubing Gui, Yong Huang, Menghan Wu, Xilai Lu, Yongming Hu, Wanping Chen

**Affiliations:** 1Key Laboratory of Artificial Micro- and Nano-Structures of Ministry of Education, School of Physics and Technology, Wuhan University, Wuhan 430072, China; guifubing@whu.edu.cn (F.G.); 2020282020113@whu.edu.cn (Y.H.); 2021282020120@whu.edu.cn (M.W.); 2021282020118@whu.edu.cn (X.L.); 2Research Institute of Wuhan University in Shenzhen, Shenzhen 518057, China; 3Hubei Key Lab. of Ferro- and Piezoelectric Materials and Devices, Faculty of Physics and Electronic Science, Hubei University, Wuhan 430062, China; huym@hubu.edu.cn

**Keywords:** CO, sensors, SnO_2_, aging, heat treatment, recover

## Abstract

A high long-term stability is crucial for room-temperature gas-sensitive metal oxide semiconductors (MOSs) to find practical applications. A series of Pd-SnO_2_ mixtures with 2, 5, and 10 wt% Pd separately were prepared from SnO_2_ and Pd powders. Through pressing and sintering, Pd-SnO_2_ composite nanoceramics have been successfully prepared from the mixtures, which show responses of 50, 100, and 60 to 0.04% CO-20% O_2_-N_2_ at room temperature for samples of 2, 5, and 10 wt% Pd, respectively. The room-temperature CO-sensing characteristics were degraded obviously after dozens of days’ aging for all samples. For samples of 5 wt% Pd, the response to CO was decreased by a factor of 4 after 20 days of aging. Fortunately, some rather mild heat treatments will quite effectively reactivate those aged samples. Heat treatment at 150 °C for 15 min in air tripled the response to CO for a 20 days-aged sample of 5 wt% Pd. It is proposed that the deposition of impurity gases in air on Pd in Pd-SnO_2_ composite nanoceramics has resulted in the observed aging, while their desorption from Pd through mild heat treatments leads to the reactivation. More studies on aging and reactivation of room-temperature gas sensitive MOSs should be conducted to achieve high long-term stability for room-temperature MOS gas sensors.

## 1. Introduction

CO is a highly dangerous gas and can be formed unintentionally in our ambient environments. Every year, there are many cases of CO gas poisoning [1]. Therefore, gas sensors that can effectively monitor CO and other dangerous gases in real time have attracted extensive attention [2].

At present, the most widely used gas sensors are metal oxide semiconductor (MOS) gas sensors, optical sensors, electrochemical sensors, and so on. These different types of gas sensors have their own advantages and disadvantages. Optical sensors are difficult to be applied in many cases due to their high cost. Electrochemical gas sensors have the drawbacks of short life and high cross sensitivity with other gases. MOS gas sensors are highly attractive with high sensitivity, long life, and low cost. However, these gas sensors can only operate at high temperatures (around 400 °C), which is energy consuming and dangerous [3,4,5,6,7,8]. For CO gas detection, the traditional metal oxides are SnO_2_, NiO, ZnO, and so on [9]. Nelli et al. prepared SnO_2_-Au thin film through rheotaxial growth and thermal oxidation technique (RGTO) and it is responsive to CO and CH_4_ at 400 °C [10]. Wang et al. invented a high temperature mixed potential CO gas sensor for on-site combustion control, and these porous NiO sensors were able to detect a low CO range of 0–100 ppm at 1000 °C [11]. Wang et al. obtained Pt/SnO_2_ nanostructures through microwave-assisted hydrothermal synthesis with strong and quick responses 100 ppm CO at 225 °C [12].

In recent years, many investigations have been conducted to develop room-temperature CO sensors based on MOSs and some remarkable results have been achieved. In 2015, Pan et al. developed an ultra-sensitive room-temperature CO sensor using zinc oxide nanocombs, whose peak sensitivities to 250 ppm and 500 ppm CO are as high as 7.22 and 8.93, respectively [13]. In 2018, Wang et al. prepared palladium(Pd)–SnO_2_ composite nanoceramics with highly attractive room-temperature CO-sensing capabilities, including high sensitivities around 15, short response time of 20 s and recovery time of 60 s for 100 ppm CO in air [14]. In 2019, it was further revealed that the formation of Pd^4^^+^ in Pd–SnO_2_ composite nanoceramics is responsible for the observed room temperature CO-sensing capability [15]. In 2020, Stanoiu et al. fabricated CO sensors from In-doped Pd-SnO_2_ powders through screen-printing and heating, which showed sensitivities around 100 to 50 ppm CO at 50 °C [16]. These contributions suggest that room-temperature MOS CO sensors are highly promising for practical applications in the near future.

Service behavior represents a basic performance for all kinds of materials. As for gas sensors, their long-term performances, including sensitivity, selectivity, response, and recovery speeds, are of vital importance for their practical applications [17]. Before room-temperature MOS gas sensors can be successfully commercialized, their long-term stability has to be systematically evaluated. For room-temperature hydrogen sensitive Pt-SnO_2_ composite nanoceramics, their room-temperature response to 1% H_2_ remained stable for about 6 months, however, their recovery time in air was dramatically increased [18]. For Pt/WO_3_ thin films, their room-temperature gasochromic hydrogen response was slightly decreased after 43 days of aging, and both the response and recovery speeds were severely decreased [19]. From these limited studies, it can be concluded that the long-term stability may represent as a major obstacle for room-temperature MOS gas sensors in their course of commercialization and should be studied systematically. 

Though several kinds of room-temperature CO sensors have been prepared and studied [13,14,15,16,20,21,22], there have been no reports on the long-term stability of room-temperature CO sensors based on MOSs in the literature up to date. In particular, an impressive room-temperature CO-sensing capability has been obtained for Pd-SnO_2_ composite nanoceramics, whose Pd content has been revealed to have an intriguing effect on the formation of the room-temperature CO-sensing capability [14,15]. As bulk materials prepared through pressing and sintering, these composite nanoceramics are more promising for practical applications than low-dimensional nanostructured MOSs in robustness and mass production. Presently, we have prepared Pd-SnO_2_ composite nanoceramics from SnO_2_ and Pd powders with a series of Pd content. All samples exhibit strong responses to CO at room temperature, and a study on the time dependence of the room-temperature CO-sensing characteristics has been conducted. Their important room-temperature CO-sensing parameters, including their response to CO, their response and recovery speeds, all degrade seriously with increasing aging time. Interestingly, the room-temperature CO-sensing characteristics of those aged samples can be mostly restored through rather mild heat treatments (e.g., 150 °C for 15 min in air). These results should be highly meaningful for the development of room-temperature CO sensors based-on MOSs with high long-term stability. To our knowledge, there have been no reports on reactivating aged room-temperature gas-sensitive MOSs through such mild heat treatments. 

## 2. Materials and Methods

### 2.1. Materials Preparation 

SnO_2_ nanoparticles (99.99%, 70 nm, Aladdin, Shanghai, China), Pd particles (99.9%, ~1 μm, Aladdin, Shanghai, China) were used as the starting materials. First, three kinds of mixtures were prepared by dispersing SnO_2_ nanoparticles and Pd particles into 30 mL deionized water at weight ratios of 90:10, 95:5, and 98:2, separately. The mixtures were stirred at 800 rpm for 4 h on a magnetic stirrer, dried in an oven at 100 °C for 15 h. The obtained dry powders were ground for 30 min, then deionized water was added as a binder and pellets with a diameter of about 10 mm and a thickness of 1.5 mm were pressed from the powders through a hydraulic press at about 3 MPa. Finally, the pellets were sintered at 950 °C in air for 2 h. A pair of rectangular Au electrodes was formed on a major surface of sintered pellets through direct current (DC) magnetron sputtering for gas-sensing measurement, as described in some previous papers [14,15].

### 2.2. CO-Sensing Measurement

A commercial gas-sensing measurement system (GRMS-215, Partulab Com., Wuhan, China) was used for CO-sensing measurement. The samples were placed in a sealed chamber (about 350 mL) with four gas inlets and one gas outlet, which are used to change the gas environment in the chamber. In the response process, specific atmospheres were introduced through mixing O_2_, N_2_, and 0.1% CO in N_2_ at some designed ratios, as shown in Figure 1. The total gas flow rate was 300 mL/min. For the recovery process, air was pumped into the chamber at a rate of 1000 mL/min. A DC voltage of 2 V was applied between the Au electrodes of the samples, and the flown electric current was measured through a Keithley 2400 Source/Meter. The room temperature was kept at 25 °C and the relative humidity (RH) in air was maintained at 50% through a commercial humidifier. 

### 2.3. Materials Characterization 

X-ray diffraction (XRD) patterns were recorded on an X-ray diffractometer (BRUKER AXS D8 ADVANCE) using Cu Kα radiation. A scanning electron microscopy (SIRION, FEI, The Netherlands) was used to analyze microstructure. Energy dispersive spectroscopy (EDS) analyses were recorded through OXFORD Aztec 250 instrument.

## 3. Results and Discussion

### 3.1. Phase and Microstructural Analysis

Figure 2 shows a representative XRD diffraction pattern, which was taken on the surface of a sintered pellet of 5 wt% Pd. It can be seen that most strong peaks are from rutile SnO_2_ phase, and some peaks from metallic Pd can also be observed. It indicates that the sintered pellets we obtained are composites of SnO_2_ and Pd. This is in agreement with a previous paper, which shows that metallic Pd is formed for the system of Pd-SnO_2_ when it is heat treated at temperatures above 900 °C [15]. Generally speaking, for these sintered pellets of 2, 5, and 10 wt% Pd, the intensity of Pd peaks increases with increasing Pd content. 

An SEM micrograph analysis for a sample of 5 wt% Pd sintered at 950 °C for 2 h is shown in Figure 3a. First of all, it should be pointed out that the microstructure is quite porous. Many pores of quite different size and shape can be observed. As a matter of fact, there were no sintering shrinkages in the diameter of the pellets, which is due to a unique sintering behavior of SnO_2_ nanoparticles [14,15]. In a previous investigation, no sintering shrinkage was observed even for pressed pellets of SnO_2_ nanoparticles sintered at 1200 °C [23]. Second, two kinds of grains with greatly different sizes can be seen in the microstructure. According to EDS analyses as shown in Figure 3b, the one around 70 nm in size is SnO_2_ grains; and the other much larger one is Pd grains. For sintered pellets of 2, 5, and 10 wt% Pd, the number of these much larger Pd grains in SEM micrograph increases with increasing Pd content. In view of the as-received SnO_2_ nanoparticles, the sintering has resulted in the formation of many grain-boundaries between SnO_2_ grains, while grain growth and densification are quite limited. Such a sintering is actually especially helpful for gas-sensing applications. 

### 3.2. Room-Temperature CO-Sensing Capabilities

For the Pd-SnO_2_ composite nanoceramics prepared by Zhu et al., only those of relatively low Pd content (≤2 wt%) and heat treated at unusually high temperatures (e.g. ≥1000 °C for samples of 2 wt% Pd) show strong responses to CO at room temperature [15]. While Pd-SnO_2_ composite nanoceramics with 2, 5, and 10 wt% Pd prepared in this study all exhibit strong responses to CO at room temperature, as shown in Figure 4. For the Pd-SnO_2_ system, it was found that metallic Pd particles are formed at high temperatures, whose surface is partially of +2 valence and partially of +4 valence. For samples prepared by Zhu et al., as obvious room-temperature CO-sensing capability can only be observed in those samples with the presence of Pd^4+^, it has been proposed that CO molecules are first chemisorbed on Pd nanoparticles at sites of Pd^4+^, and through spill-over effect they are moved to and chemisorbed on SnO_2_ with their electrons donated to SnO_2_ [15]. On the other hand, as both relatively low Pd content (≤2 wt%) and heat treatment at unusually high temperatures are necessary for the formation of Pd^4+^, it is clear that SnO_2_ has a vital role in the oxidation of Pd, namely the formation of Pd^4+^ at high temperatures [15]. This role of SnO_2_ in the oxidation of Pd must be depressed when too much Pd is present. For the samples prepared by Zhu et al., metallic Pd was formed through the following replacement reaction:(1)$$\mathrm{Zn}+{\mathrm{Pd}}^{2+}\to {\mathrm{Zn}}^{2+}+\mathrm{Pd},\;$$
and the particles were only a few nanometers in size. So, for the same Pd content, there was much larger contact area between Pd and SnO_2_ and much stronger depression effect of Pd for the samples prepared by Zhu et al. Pd grains are much larger in our samples and Pd^4+^ must have been formed even in samples of 10 wt% Pd. It is thus reasonable that room-temperature CO-sensing capabilities have been observed for all samples prepared in our study.

For gas sensors, the response *S* is usually defined as *S* = *R*_a_/*R*_g_, where *R*_a_ and *R*_g_ represent the resistances of the sensors in air and in the test gas, respectively. The time taken by the sensors to reach 90% of the total resistance change in the response (recovery) process is defined as the response (recovery) time. From Figure 4, the samples of 2, 5, and 10 wt% Pd show responses of 50, 100, and 60 to 0.04% CO-20% O_2_-N_2_ at room temperature, respectively, with response times of 56, 44, and 57 s, respectively, and recovery times of 53, 39, and 41 s, respectively. It is clear that these three kinds of samples all show strong and stable responses to CO at room temperature, with both especially quick response and recovery speeds. As a matter of fact, similar results were also obtained for Pd-SnO_2_ composite nanoceramics in some previous investigations [14,15]. It is worthy to note that SnO_2_-based gas sensors usually show responses to multiple reducing gases. While a Pd–SnO_2_ composite nanoceramic sample showed a representative response of 2 to 0.08% H_2_-20% O_2_-N_2_ at room temperature [14], a Pt–SnO_2_ sample showed a response of 12 to 0.06% H_2_-20% O_2_-N_2_ under the same condition [18], which indicates that Pd–SnO_2_ composite nanoceramics have a better selectivity for CO against H_2_ than Pt–SnO_2_ composite nanoceramics. Obviously, Pd-SnO_2_ composite nanoceramics are especially promising to be applied as room-temperature CO sensors in future. Samples with 5 wt% Pd show the strongest response and our further studies are focused on samples of 5 wt% Pd. 

### 3.3. Time Dependence of Room-Temperature CO-Sensing Characteristics of Pd-SnO_2_ Composite Nanoceramics

To study their long-term stability, the samples prepared in this study were kept at room temperature for dozens of days, and their room-temperature CO-sensing characteristics were measured repeatedly with some time intervals.

Unfortunately, all samples were found to degrade obviously in their room-temperature CO-sensing characteristics after dozens of days’ aging. As an example, the results measured for a sample of 5 wt% Pd as-sintered, 12 and 20 days aged, separately, are shown in Figure 5. After 20 days of aging, the response to 0.04% CO-20% O_2_-N_2_ was 27, which was almost only one-fourth of that as-sintered, and the response and recovery times were increased to 55 and 202 s, respectively. It is reasonable to assume that the sample will eventually show no response to CO at room temperature when the aging time is long enough.

Such an aging behavior is actually very similar to that reported for room-temperature hydrogen-sensitive MOSs [18,19]. In the future, there must be more and more reports on aging behavior of various kinds of room-temperature gas-sensitive MOSs when they are becoming more and more promising for practical applications. Obviously, if it is not overcome in some way, aging behavior will prevent those room-temperature gas sensors from commercialization, or greatly decrease their service time. 

### 3.4. Mild Heat Treatment for Aged Room-Temperature CO Sensitive Pd-SnO_2_ Composite Nanoceramics

Heat treatment is widely adopted to reactivate aged MOS gas sensors. For Pt/WO_3_ thin films room-temperature gasochromic hydrogen sensors, heat treatment at 500 °C for 1 h restored the response and recovery speeds completely [19], which were both severely decreased after 43 days’ aging. For commercial SnO_2_ thick-film gas sensors, several days of inactivity or aging usually make them “dormant”. A significant pre-heating up to 300–700 °C is needed to reactivate them [24,25]. Obviously, these results clearly indicate that heat treatment is a very effective method for the reactivation of aged MOS gas sensors. 

For room-temperature gas sensors, it is meaningful to heat-treat them, if inevitable, at temperatures as low as possible. Aged Pd-SnO_2_ composite nanoceramics prepared in this study had therefore been heat treated at especially low temperatures and some very encouraging results had been obtained. As shown in Figure 5, the room-temperature CO-sensing capability of a sample of 5 wt% Pd was seriously degraded after 20 days’ aging. It was heated in air for 15 min at 100 °C, 150 °C, and 200 °C, separately, and its room-temperature CO-sensing capability was measured immediately after every heat treatment. It is quite surprising that the heat treatment of 100 °C resulted in an obvious improvement to the room-temperature CO-sensing capability, as shown in Figure 6. For the heat treatment of 150 °C, a better improvement was obtained. The response to 0.04% CO-20% O_2_-N_2_ was 75, which was almost three times of that 20 days-aged. However, when the sample was heat treated at 200 °C, its room-temperature CO-sensing capability was degraded obviously once again, with a response of only 23 to 0.04% CO-20% O_2_-N_2_, as shown in Figure 6. It seems that there exists an optimal temperature for heat treatment to reactivate aged Pd-SnO_2_ composite nanoceramics. It should be pointed out that even the heat treatment of 150 °C did not lead to a full reactivation. The recovery speed was still much lower than that of the as-sintered sample. More investigations should be conducted to achieve as full as possible reactivation. 

### 3.5. Discussion on Aging Origin and Strategies for Developing Stable Room-Temperature Gas Sensors Based-on MOSs

Generally speaking, materials’ aging behavior results from types of changes: one is internal evolutions in materials, and the other is the reactions from environmental factors. Pd and SnO_2_ both are very simple and stable at room temperature. The observed aging behavior should not be caused by any internal evolutions inside them at room temperature. On the other hand, there are water and many purity gases in air, among which more than 900 kinds of volatile organic compounds have been identified. The reaction of water with many materials at room temperature has been well studied [26,27]. Recently, Fei et al. found that for TiO_2_ air cleaning photocatalyst, impurity gases in air, such as ammonia, sulfur compounds, and carbon compounds, can deposit and block the surface-active sites of TiO_2_, which is responsible for a rapid deactivation of TiO_2_ catalyst [28]. Similarly, it is reasonable to assume that some impurity gases in air also deposit and block the surface-active sites of Pd in Pd-SnO_2_ composite nanoceramics, which leads to a decrease in the response to CO. The interaction between the impurity gases and Pd must be very weak and rather mild heat treatments are able to desorb them from Pd. In this way the obvious reactivations by the mild heat treatments can be well explained.

While degradation over long time aging is difficult to avoid for many room-temperature gas sensitive MOSs, our findings in this study actually suggest a relatively simple strategy for developing stable room-temperature gas sensors based on MOSs. With a heater and a circuit, room-temperature gas sensitive MOSs can be reactivated through a short and mild heat treatment periodically. In this way, MOSs not only can maintain highly attractive room-temperature gas-sensing capabilities for longer periods of time, but also are still much safer and more cost-effective than high-temperature MOS gas sensors. Further studies on aging and reactivation are highly expected for the development of room-temperature MOS gas sensors with high long-term stability. 

## 4. Conclusions

Pd-SnO_2_ composite nanoceramics have been prepared from SnO_2_ and Pd powders through pressing and sintering, which show strong responses of 50, 100, and 60 to 0.04% CO-20% O_2_-N_2_ at room temperature for samples of 2, 5, and 10 wt% Pd, respectively. An obvious aging behavior was observed for all samples, with the response to CO decreased by a factor of 4 after 20 days of aging for samples of 5 wt% Pd. However, those aged samples can be effectively reactivated through rather mild heat treatments. Heat treatment at 150 °C for 15 min in air tripled the response to CO for a 20-days-aged sample of 5 wt% Pd. These findings suggest that impure gases in air must have deposited on Pd and blocked the surface-active sites of Pd in Pd-SnO_2_ composite nanoceramics, which resulted in the observed aging. On the other hand, those impure gases can be removed through rather mild heat treatments and the aged samples can be reactivated. A detailed understanding of aging and reactivation behaviors of room-temperature gas sensitive MOSs is highly desirable for developing room-temperature MOS gas sensors with high long-term stability.

## Figures and Tables

**Figure 1 materials-15-01367-f001:**
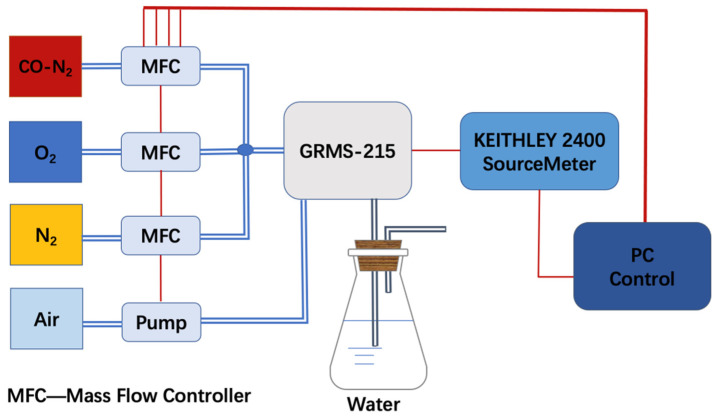
A scheme for CO-sensing measurement set-up.

**Figure 2 materials-15-01367-f002:**
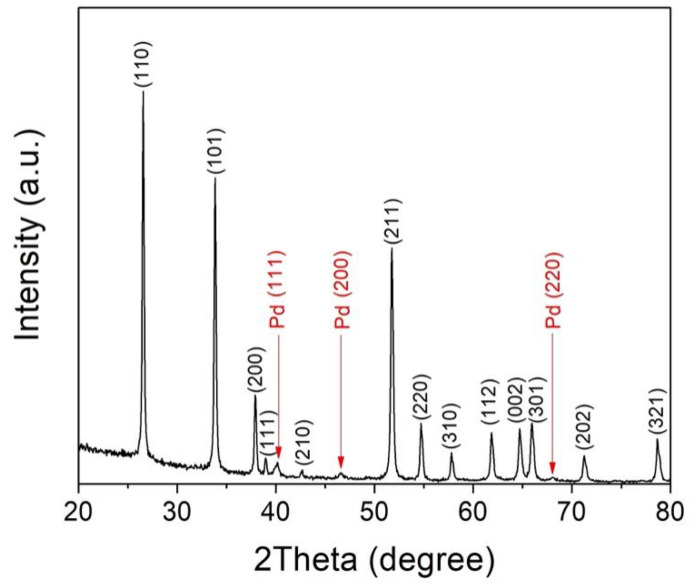
X-ray diffraction pattern taken for the surface of a pellet of 5 wt% Pd after being sintered at 950 °C for 2 h in air.

**Figure 3 materials-15-01367-f003:**
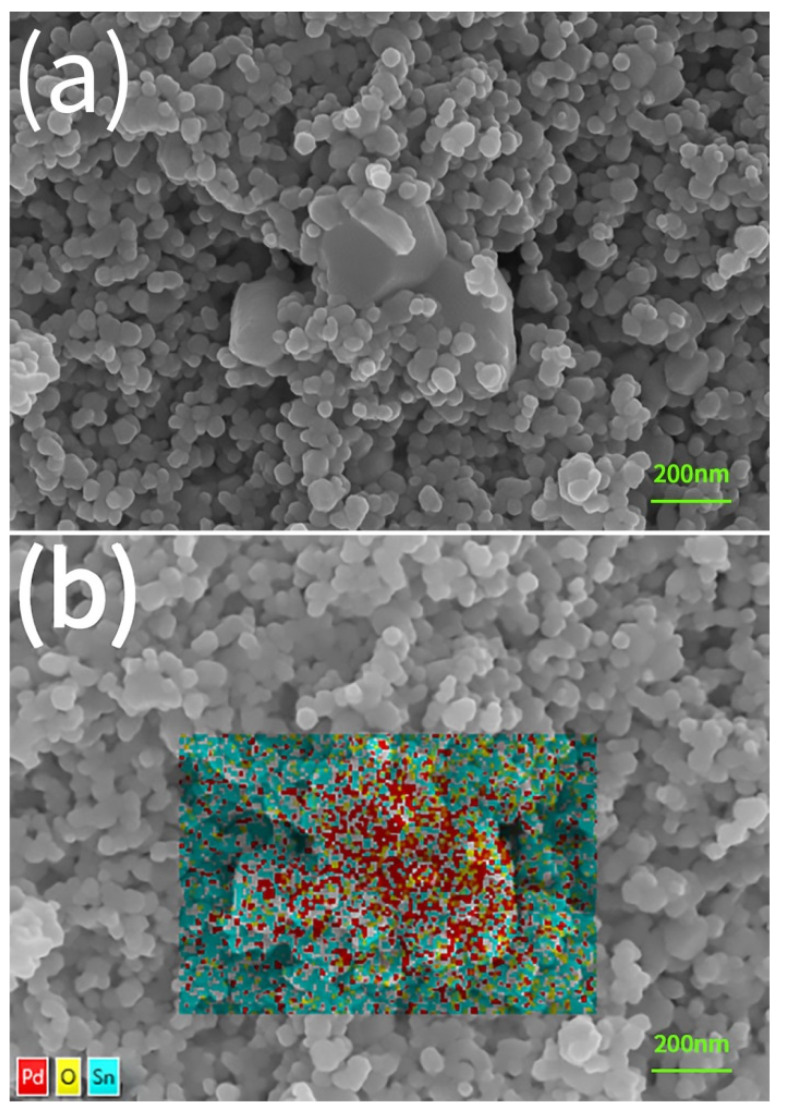
SEM Micrograph analyses for a Pd-SnO_2_ nano-ceramic sample of 5 wt% Pd sintered at 950 °C for 2 h in air: (**a**) an SEM micrograph; (**b**) an SEM micrograph with EDS analysis covering its large grains in the center.

**Figure 4 materials-15-01367-f004:**
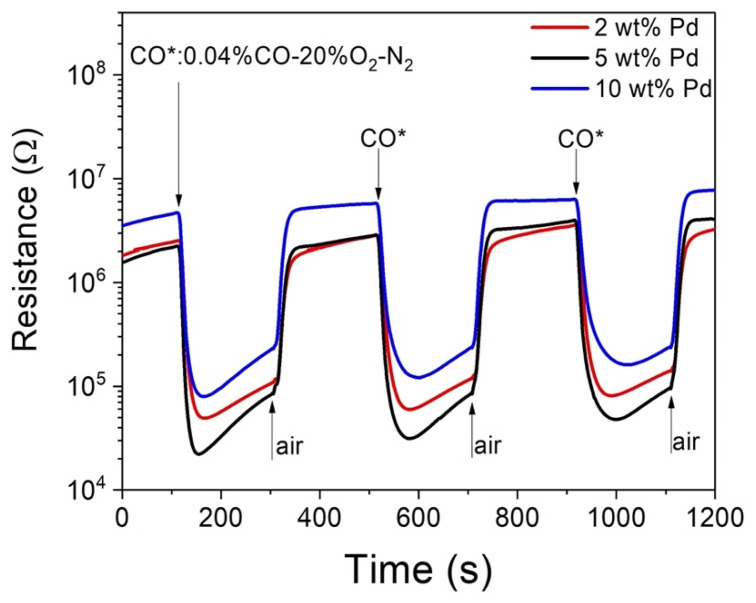
Room-temperature response to 0.04% CO-20% O_2_-N_2_ and recovery in air for Pd-SnO_2_ nano-ceramic samples with 2, 5, and 10 wt% Pd, separately.

**Figure 5 materials-15-01367-f005:**
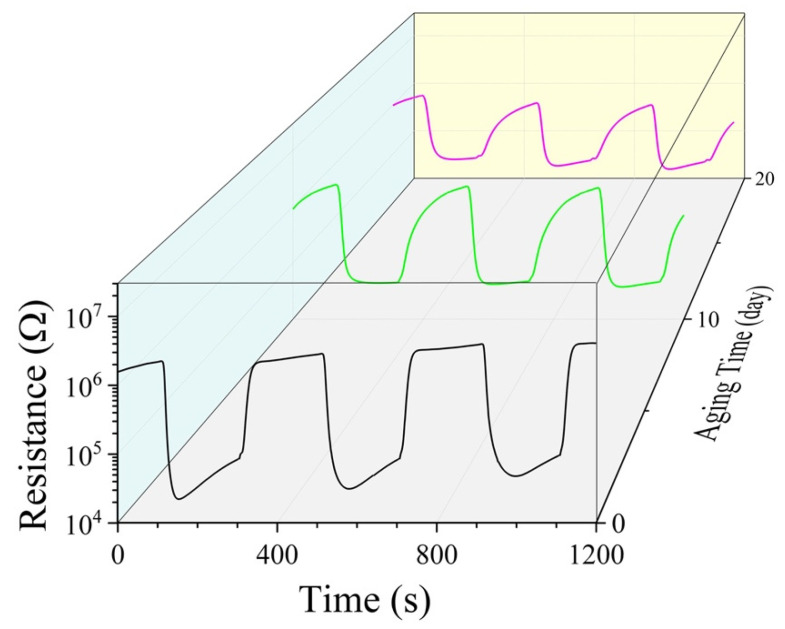
Resistance responses to 0.04% CO-20% O_2_-N_2_ and recovery in air of 50 RH at room-temperature for a sample of 5 wt% Pd as-sintered, and after aging of 12 and 20 days, separately.

**Figure 6 materials-15-01367-f006:**
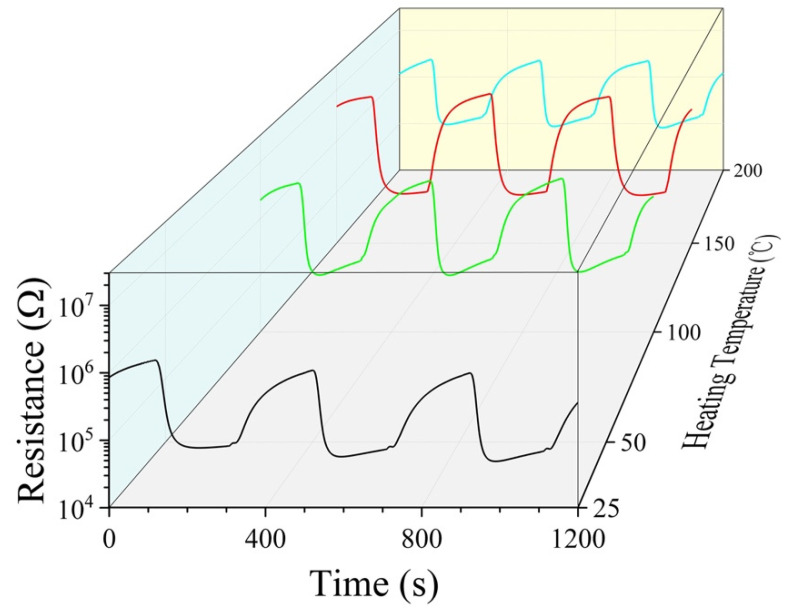
Resistance responses to 0.04% CO-20% O_2_-N_2_ and recovery in air of 50 RH at room-temperature for a sample of 5 wt% Pd after 20 days of aging, and then being heat treated at a series of temperatures (100, 150, and 200 °C), separately.

## Data Availability

Not applicable.

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
