# Peer review of "Aging Behavior and Heat Treatment for Room-Temperature CO-Sensitive Pd-SnO_2_ Composite Nanoceramics"

_materials, 2022, doi:10.3390/ma15041367_

Round 1
Reviewer 1 Report
I read your manuscript entitled " Aging behavior and heat-treatment for room-temperature CO 2 sensitive Pd-SnO2 composite nanoceramics", my comments are as follows:
1- I think aging days should last to 60 days....
2- Selectivity studies are necessary.
3- TLV of CO gas should be mentioned.
4- More SEM /TEM images are necessary.
5- The sensing mechanism should be discussed.
6- The effect of water vapor on the aging and response should be studied
7- Low concentrations of CO should be tested.
Reviewer 2 Report
In this paper, the authors reported Aging behavior and heat-treatment for room-temperature CO sensitive Pd-SnO2 composite nanoceramics. In general, the results of CO gas sensing and composite characterization in the manuscript are well-organized and acceptable for publishing. For improving a manuscript, it is recommended to address the following comments:
- The abstract should be clearer to include the most important results obtained from this work and conclusion also.
- In the introduction, the authors should focus on the benefits and the reason for choosing Pd-SnO2 as functional material for gas sensing should be included and its advantages compared to other MOS.
- In the experimental, the purity of chemicals should be added.
- In the experimental, the reference of Pd-SnO2 preparation should be added.
- In the experimental, how the authors obtained a uniform surface from pellets to make the gold electrodes? It is recommended to add a scheme for synthesis and sensor preparation.
- In the experimental, How the authors adjusted the RH at 50%
- In results, the reference of sensitivity equation should be added and the authors should clear the reference gas is air or nitrogen.
- In results (line 156), the response values of Pd Concs should be mentioned.
- The comparison of current work with literature is very important and should be added.
- The selectivity test with other gases is recommended.
- The authors should calculate the response and recovery time for each sensor when the Pd changed.
- The mechanism of gas sensing should be described.
Reviewer 3 Report
The authors have investigated aging of Pd-SnO2 nanocomposite samples with 2, 5 and 10%Pd and their improvement with heat treatment. Overall the paper is too general and needs to contain more actual data. First, the nancomposites need to be completely characterized. XRD of all three sample compositions need to be given. SEM images need to be given for all three samples compositions and compared. The authors claim to have done EDS analysis, yet no results are given. There is no proof that the Pd is +4 and additional XPS analysis needs to be done to prove this. The authors claim to have obtained a very strong response for their nanocomposites. They need to compare their results with available literature data in terms of the gas mixture and response and recovery time and sensitivity (gas mixture amount). The images given in the figures are not enough. Concerning the heat treatment, these results also need to be quantified and compared and their relevance emphasized with actual data values.
Reviewer 4 Report
Aging behavior and heat-treatment for room-temperature CO sensitive Pd-SnO2 composite nanoceramics
The work is devoted to long-term stability of Pd-SnO2 composite nanoceramics at room-temperature CO sensing, which is an important issue in terms of application and commercialization of these promising materials. Some comments and suggestions you can find below.
Introduction
The preparation of composites with different amount of paladium was not substantiated.
Provide comparative discussion of the current research and works from last years devoted to room temperature CO sensing composites (for example, Stanoiu, A., Ghica, C., Somacescu, S., Kuncser, A. C., Vlaicu, A. M., Mercioniu, I. F., ... & Simion, C. E. (2020). Low temperature CO sensing under infield conditions with in doped Pd/SnO2. Sensors and Actuators B: Chemical, 308, 127717).
There are similar works with Pt composites. What is the main difference? (Wang, Q., Bao, L., Cao, Z., Li, C., Li, X., Liu, F., ... & Lu, G. (2020). Microwave-assisted hydrothermal synthesis of Pt/SnO2 gas sensor for CO detection. Chinese Chemical Letters, 31(8), 2029-2032.).
Provide the sensing mechanism shortly to instill confidence that this composite should operate long-life.
Materials and methods
Was the sintering optimized or already known parameters form other works were used? Did you use the same parameters for all the samples?
Detail SEM sample preparation, as they look like to not well-sintered bulks.
Were XRD patterns detected from the surface of the sample or fracture? Did you detect any difference in phase composition from surface or fracture? What is the modification of initial SnO2 nanoparticles (rutile)?
Results
Explain the rows 114-116. What did you mean by ’ metallic Pd is formed’?
Line 126, ’ Such a sintering is actually especially helpful for gas-sensing applications.’ Any proof for that?
Figure 2, doesn’t look like sintered sample, but pressed and heated. Present SEM images for all composites.
Please provide clear comparison of your work and Zhu et all [14] for 2wt% Pd containing composites. Why your samples behave better?
Clarify paragraph from line 139 to 146.
’SnO2 has a vital role in the formation of Pd4+ at high temperatures’….do you compare here samples sintered at 950 and 1000oC?
’Metallic Pd was formed through a replacement reaction’… write the reaction equation
If you would like to comment on the differences of your work and others, first, you should explain CO sensing mechanism. How Pd-SnO2 provides high sensing behaviour to CO at room temperature? What is the main interaction here and is it reversible?
The only difference to your work and reference [14] could be attributed to porosity and particle size, but it should be presented properly. Provide relative density measurements.
Your samples showed responses of 50, 100, and 60 to 0.04% CO-20% O2-N2 at room temperature for 2, 155 5, and 10 wt% Pd, respectively. Did you compare them with others? Are they better in terms of short response?
It was not clear why 100oC was good for heat treatment and 150 or 200oC is not. What is happening at 150oC that degrades the sensing properties of composite? And why the sample with 5wt%Pd was selected? How the other two composites behave at similar conditions and why?
11. 12. 2021
Reviewer 5 Report
The manuscript entitled “Aging behavior and heat-treatment for room-temperature CO sensitive Pd-SnO2 composite nanoceramics” authored by Fubing Gui, Yong Huang, Menghan Wu, Xilai Lu, Yongming Hu, and Wanping Chen has been reviewed.
This paper investigates the high long-term stability of Pd-SnO2 based room-temperature CO gas sensitive MOSs, their degradation as a function of time and the effective reactivation of the MOSs after mild heat treatments. The authors have prepared various compositions of Pd-SnO2 composite nanoceramics with strong responses to CO and studied the RT sensing characteristics as a function of time, aging behavior and a method to restore the sensing characteristics through mild heat-treatments of the aged samples in air. These results should be highly meaningful for the development of room-temperature CO sensors based-on MOSs with high long-term stability.
The scientific content of the manuscript is very significant to understand the aging and reactivation of room-temperature gas sensitive MOSs nano composite based gas sensors to achieve high long-term room-temperature stability. The paper is very interesting and reports the solution for a very critical issue found in gas sensors and worth publishable.
The manuscript is very well written with suitable references but lack some more information described below to make it meaningful. Therefore, a minor revision of the manuscript is required before publishing it.
The scientific content of the paper is novel and adequate for publication after making the following corrections
Q1. Define “Pd” and indicate it with symbol (Pd) in the text.
Q2. Page1, Line-44-45 “In recent years, many investigations have been devoted to developing room-temperature CO sensors based-on MOSs and some remarkable results have been achieved.”
Please correct the grammar.
Q3. This manuscript is lacking pictures or schematics of the gas sensing experiments conducted in this study. Please provide a schematic representation or a photograph of the CO Sensing Measurement set up.
Q4. P3, L-97-98, “Specific atmospheres were introduced through mixing O2, N2, and 0.1% CO in N2 at some designed ratios.”
How accurate was your gas mixtures? What ratios you designed in mixing O2, N2, and 0.1% CO gases at some designed ratios in N2? Are they already mixed gases? Or physical mixing? Please indicate the gas mixing ratios you designed on a Table.
Q5. P3, L-106-107, “Scanning Electron Microscopy was used to analyze microstructural performed through SIRION TMP”.
Please correct the English grammar of the sentence. Please explain how you made the sample preparation for SEM and what analysis your conduct through SIRION TMP?
Q6. P3, L-109, Please correct the side heading “Results and Discussions” as “Results and Discussion”
Q7. P3, L-111-112, “The authors mentioned that three compositions of Pd-SnO2 mixtures with 2, 5, and 10 wt% Pd were prepared in this study. However, the X-ray phase analysis only give a representative analysis of 5% Pd-SnO2 composite? What is the effect of Pd content on the phase analysis of other composite samples? Please briefly discuss the analysis results of other compositions in one or two sentences.
Q8. P3, L-120-121, “there were no sintering shrinkages in the diameter of the pellets, which is due to a unique sintering behavior of SnO2 nanoparticles”.
What are the unique sintering characteristics of SnO2 you mentioned? Please explain briefly. Your SEM pictures shows that your sample is porous. Did you measure the density of the sintered samples?
Q9. P4, L-130, In the characterization section, you have mentioned that you have done EDS analysis of your nanocomposite samples. Is it possible to differentiate the SnO2 and Pd grains in your SEM picture? If yes, please label it on the SEM micrograph.
Q10. P4, L131-132, In the Figure 2 caption “Figure 2. SEM Micrograph analyses for a Pd-SnO2 Nano-ceramic sample of 5 wt% Pd sintered at 131 950 ℃ for 2 h in air.”
Please change the Nano-ceramic to “nano-ceramic”.
Q11. P4, L140-147, “For the samples prepared by Zhu et al., metallic Pd was formed through a replacement reaction and was only a few nano meters in size. So for the same Pd content, there was much larger contact area between Pd and SnO2 and much stronger depression effect of Pd for the samples prepared by Zhu et al. Pd grains are much larger in our samples and Pd4+ must have been formed even in samples of 10 wt% Pd. It is thus reasonable that room-temperature CO sensing capabilities have been observed for all samples prepared in our study.”
This part of the discussion is not clear. Briefly explain what type of replacement reaction occurred in Zhu et al.’s samples? Also what do you mean by the “depression effect of Pd” Please clarify?
Q12. P8, L254-257. “A detailed understanding of aging and reactivation of room-temperature gas sensitive MOSs should be a key to achieving high long-term stability for room-temperature MOS gas sensors”.
Please correct the English grammar of the sentence.
Round 2
Reviewer 1 Report
To approve the presence of Pd in Fig. 2, EDS analysis is necessary...
Author Response
This is an important suggestion. We have hurried to do more SEM analyses to obtain an original SEM micrograph and an SEM micrograph with EDS data that can match each other well, as shown in the revised manuscript.
Reviewer 3 Report
The authors have made some improvements to their work, but only marginal. I am not satisfied with the replies to the my comments on the original paper. The authors have not added any more data, only a general sentence when more XRD data, SEM data, XPS measurements were asked for. Instead they referred to their previous work, so this paper is only a continuation of previous work focusing only on aging and 5 wt/% Pd as relevant. I suggest the authors rewrite their work to focus only on that in the form of a communication rather that paper.
Author Response
We have conducted investigations on room-temperature gas sensitive metal oxide semiconductors (MOSs) for several years. In particular, we have successfully developed bulk MOSs with highly impressive room-temperature H2 and CO sensing capabilities, separately. As bulk materials, they are especially promising for practical applications with regards to their mechanical robustness and mass production. At this stage, however, the importance of long-term stability, or aging, has come into our attention. We have found that this problem has been quite neglected up to date in the literature. In our opinion, it will be the last major challenge that room-temperature MOSs must overcome before they can find practical applications. We believe that this paper should be able to attract much attention from researchers in this field and stimulate many investigations on this important phenomenon.
Reviewer 4 Report
The novelty of the work should be clearly stated. The scientific concept behind the sensing behaviour observed at aging should be properly explained.
Author Response
The novelty of the work should be clearly stated.
Reply: The following sentence has been added to show the novelty of this paper:
To our knowledge, there have been no reports on reactivating aged room-temperature gas sensitive MOSs through such mild heat-treatments.
The scientific concept behind the sensing behaviour observed at aging should be properly explained.
Reply: We are sorry we cannot quite understand this suggestion. We have thus added the following sentence with a general meaning:
Service behavior represents a basic performance for all kinds of materials.